# Comparison of Culturing and Metabarcoding Methods to Describe the Fungal Endophytic Assemblage of *Brachypodium rupestre* Growing in a Range of Anthropized Disturbance Regimes

**DOI:** 10.3390/biology10121246

**Published:** 2021-11-29

**Authors:** María Durán, Leticia San Emeterio, Rosa Maria Canals

**Affiliations:** Department of Agricultural Engineering, Biotechnology and Food, Institute on Innovation and Sustainable Development in Food Chain (IS-FOOD), Public University of Navarre (UPNA), Arrosadia Campus, 31006 Pamplona, Spain; leticia.sanemeterio@unavarra.es (L.S.E.); rmcanals@unavarra.es (R.M.C.)

**Keywords:** *Brachypodium rupestre*, mycobiome, fire, grazing, metabarcoding, culturing

## Abstract

**Simple Summary:**

The richness (number of species) of the fungi kingdom is estimated at 1.5 million species, but the vast majority remains unknown. Many of them inhabit plants—the so-called fungal endophytes—and may establish different types of interactions with their host plant. Fungal endophytes have been traditionally studied by letting them grow in appropriate culturing media in petri dishes, but novel massive DNA sequencing techniques which do not require a cultivation step (metabarcoding) are gaining ground. Both techniques were applied and compared to characterize the mycobiome of plants of a tall grass (*Brachypodium rupestre*) growing in high-mountain grasslands with different plant diversity (low and high). The two methods showed similar trends comparing endophyte richness between plant tissue types (root > rhizome > shoot) and between grasslands (low-diversity > high-diversity). However, the metabarcoding identified almost six times more endophyte species than the culturing although the most isolated fungal species via culturing, *Omnidemptus graminis,* was not recognized via metabarcoding. We conclude that the complementation of both techniques is still the best option to obtain a complete characterization of the fungal endophytic assemblage of the plant species.

**Abstract:**

Fungal endophytes develop inside plants without visible external signs, and they may confer adaptive advantages to their hosts. Culturing methods have been traditionally used to recognize the fungal endophytic assemblage, but novel metabarcoding techniques are being increasingly applied. This study aims to characterize the fungal endophytic assemblage in shoots, rhizomes and roots of the tall grass *Brachypodium rupestre* growing in a large area of natural grasslands with a continuum of anthropized disturbance regimes. Seven out of 88 taxa identified via metabarcoding accounted for 81.2% of the reads (Helotiaceae, *Lachnum* sp. A, *Albotricha* sp. A, Helotiales A, Agaricales A, *Mycena* sp. and Mollisiaceae C), revealing a small group of abundant endophytes and a large group of rare species. Although both methods detected the same trends in richness and fungal diversity among the tissues (root > rhizome > shoot) and grasslands (low-diversity > high-diversity grasslands), the metabarcoding tool identified 5.8 times more taxa than the traditional culturing method (15 taxa) but, surprisingly, failed to sequence the most isolated endophyte on plates, *Omnidemptus graminis*. Since both methods are still subject to important constraints, both are required to obtain a complete characterization of the fungal endophytic assemblage of the plant species.

## 1. Introduction

The study of microorganisms in their natural environment is a recent branch of research compared to microbial investigations undertaken in disciplines such as medicine and agronomy, with high impact on human health and development [1,2]. Nowadays, microbial ecology, i.e., their diversity in nature, their response to prevailing and future environmental conditions, the associations they establish with plants and the complex network of interactions and functions they are involved in, are gaining ground in ecological research [3,4,5].

One example involves examining the associations that endophytic fungi establish with plants. These associations were first studied in agronomic grasses [6,7,8] and the research has extended to natural plant communities in recent decades [9,10,11]. Scientific literature has shown that these hidden associations are ubiquitous in nature and that all plants harbor an endophyte assemblage that delivers different functions and constitutes a collective and complex holobiont [12].

Nowadays, two techniques, culturing and metabarcoding, are used for the determination of fungal endophyte assemblages [13]. The protocols of culturable techniques have a longer record and have been implemented in many laboratories [14]. In this method, important constraints include the possibility that some fungal species are unculturable on artificial medium and the accumulation of inaccuracies and errors due to different sterilization times, diverse species growth rates and the presence of surface contaminants [15]. Metabarcoding techniques (culture-independent) [16], despite appearing very promising, still remain costly and lack a complete repository of sequences with taxonomic identification, a task which is under way [17,18]. In the latter, the potential for providing quantitative data based on the proportion of read sequences makes it a very powerful ecological tool [19,20].

The genus *Brachypodium* encompasses several perennial tall grasses, native to European calcareous grasslands, which have been expanding aggressively in the last decades due to the global change conditions (*B. pinnatum*, *B. genuense* and *B. rupestre)* [21,22,23,24,25]. This tall grass expansion causes a decline of the biodiversity of the natural grasslands and also has an impact on the ecosystem service of provisioning [26]. The competitive strategies of this group of species that explain the expansive process is a matter of interest [27,28,29,30,31,32,33], as it is the study of the mycobiome that may help to understand these advantages. To date, the research in the *Brachypodium* genus has focused on the systemic fungi of the Clavicipitaceae family hosted by *B. sylvaticum* [34,35], *B. phoenicoides* [36,37] and *B. pinnatum* [38]. Only a previous study of our research team has characterized the systemic and non-systemic mycobiome of *Brachypodium rupestre* under a gradient of grazing and fire disturbances using culturable techniques [39].

The aim of this research is to provide a characterization of the endophytic mycobiome of the tall grass species *Brachypodium rupestre* and to compare culture and metabarcoding techniques applied to conditions with restricted sampling effort due to the high cost of the novel technique. The comparison includes the aboveground (shoot) and the underground (rhizome and root) component of a set of *B. rupestre* individuals growing in the same region but subjected to different levels of anthropic disturbance (grasslands with different regimes of grazing and prescribed burning and, consequently, encompassing a different plant community composition). Through this range of regional variation, and considering different tissues and different environmental drivers, we are interested in determining the capacity of the two methods to identify and characterize the fungal endophyte assemblage of *B. rupestre*.

## 2. Materials and Methods

### 2.1. The Study Area

The Aezkoa valley (Navarra county, Spain) is the westernmost valley of the southern Pyrenees (42.53–43.3′ N, 1.8–1.17′ W) (Figure 1d). The climate is snowy and cold in winter, and mild and foggy in summer. The annual temperature averages 9.3 °C and the accumulated precipitation reaches 1856 mm per year (Irabia climatic station, http://meteo.navarra.es accessed on 17 October 2021). The landscape is a mosaic of forests (e.g., *Fagus sylvatica*, *Abies alba*), shrubland communities (e.g., *Erica* spp., *Ulex gallii*) and grasslands. The area of study is part of the Special Area of Conservation (SAC) Roncesvalles-Selva de Irati (code ES0000126; Figure 1f) and is located in the north of the valley. High-altitude grasslands (800–1400 m asl) comprise diverse communities of perennial grasses (*Festuca* gr. *rubra*, *Agrostis capillaris*, *Brachypodium rupestre*, *Danthonia decumbens*), forbs (*Achillea millefolium*, *Potentilla erecta*, *Gallium saxatile*) and legumes (*Trifolium repens*, *Lotus corniculatus*). Sandstones and calcareous clays dominate the substrate, upon which develop acidic, deep and organic soils, with clay-loamy and loamy textures.

Depending on the grazing pressure of the livestock during the summer months, farmers schedule different types of burnings to control the build-up of litter and resprouting of woody species. As a result, traditional (bush-to-bush) burnings applied every 6–7 years coexist with more intense fire regimes, applied across the whole surface every 1–2 years in the less grazed areas. The regional plant community composition reflects the dominant grazing/burning regime, which leads to a mosaic of high-diversity grasslands (more grazed, less burned) and low-diversity grasslands highly dominated by *B. rupestre* (less grazed, more burned). Based on previous floristic surveys undertaken in the area [26], we selected two representative locations according to the percentage of *B. rupestre* cover. A low-diversity grassland (LD) in Arpea, with a dominant cover of *B. rupestre* up to 80%, and a high-diversity grassland (HD) located in Urkulu, with a *B. rupestre* cover lower than 25% (Table 1).

### 2.2. Plant Sampling

In summer 2018, a total of 10 turfs of *B. rupestre* were collected (turfs included shoots, rhizomes and roots surrounded by soil) from the two locations (Figure 1f). The distance between turf samples was ca 150 m to avoid collecting clonal individuals. Turfs were transported to the UPNA laboratory and processed in the following days.

One *B. rupestre* plant with high biomass was selected from each turf. Tissues were separated (shoots, rhizomes and roots) and cut into fragments of ca 2 cm, surface-disinfected via immersion in a solution of 20% commercial bleach (1% active chlorine) containing 0.02% Tween 80 (v:v) for 10 min and finally rinsed with sterile water. The rhizome and root fragments were also treated with an aqueous solution of 70% ethanol for 30 s. Thirty fragments (10 shoots, 10 rhizomes and 10 roots) assigned to the metabarcoding method were ground using a pestle with liquid nitrogen and preserved at −20 °C until shipment.

### 2.3. Isolation and Identification of Fungi Using the Culturing Method

We plated 300 tissue fragments of *B. rupestre* onto 30 culture media plates (10 fragments/tissue/plate, 90 mm diameter), containing PDA medium (potato dextrose agar) with chloramphenicol (200 mg/L). Dishes with tissue fragments were kept at room temperature and ambient light and checked daily for 4 weeks. Any emerging mycelium was transferred and individually isolated in a new mini petri dish (60 mm diameter). Isolates with the same morphological characteristics (colony color, exudates, growth type and general appearance) were grouped into morphotypes, and at least one of them was genotyped for taxonomic analysis.

A small amount of mycelium was collected and its DNA extracted using a Phire Plant Direct PCR Kit (Thermo Fisher Scientific). The complete ITS region (ITS1-5.8S-ITS2) was amplified using ITS4 and ITS5 primers [40]. The amplification cycles followed were: 98 °C for 5 min, 95 °C for 5 s (35 repeated cycles), 54 °C for 5 s, 72 °C for 20 s and a final phase of 72 °C for 1 min. PCR amplicons were purified (Favor PrepTM Plant Genomic DNA Extraction Mini Kit, Favorgen) and sequenced using the Sanger method, copying single-stranded DNA, at STABVIDA enterprise. The returned DNA sequences were grouped using the CD_HIT program at 97% identity threshold [41,42], considering the clustered sequences to represent the same taxon. A representative sequence of each cluster was selected and contrasted to the closest match of the ITS region from fungal types at the National Centre for Biotechnology Information (NCBI) using the BLAST algorithm [43]. The database UNITE was also interrogated for sequences.

### 2.4. Metabarcoding Analysis and Taxonomic Assignment

A total of 30 samples was sent to AllGenetics services for metabarcoding analysis. The DNA of samples was isolated using a Dneasy PowerSoil DNA isolation kit (Qiagen, Hilden, Germany), and the complete ITS2 region was amplified using the primers ITS86F and ITS4 [40,44], to which the Illumina sequencing primer sequences were attached to their 5’ ends. The PCR cycle consisted of an initial denaturation at 95 °C for 5 min, followed by 35 cycles of 95 °C for 30 s, 49 °C for 30 s, 72 °C for 30 s and a final extension step at 72 °C for 10 min. The index sequences required for multiplexing libraries were attached in a second PCR with the same conditions but only 5 cycles and 60 °C as the annealing temperature. Libraries were purified using Mag-Bind RXNPure Plus magnetic beads (Omega Biotek, Norcross, GA, USA), pooled in equimolar amounts and sequenced in a MiSeq PE300 run (Illumina, San Diego, CA, USA).

The Illumina raw files R1 (forward) and R2 (reverse) reads were trimmed and checked using the software FastQC (www.bioinformatics.babraham.ac.uk accessed on 17 October 2021). FLASH2 was used to merge reads and CUTADAPT software 1.3 to remove sequences that did not contain the PCR primers and those shorter than 100 nucleotides [45,46]. The sequences were filtered by quality using Qiime v1.9.1. and the FASTA file was processed using VSEARCH [47]. Sequences were dereplicated, sorted and clustered at a similarity threshold of 100%. Artefacts were detected and filtered using the UCHIME algorithm implemented in VSEARCH [48]. Sequences were then assigned to OTUs, and those occurring at a frequency below 0.005% in the whole dataset were removed. In the same way as the sequences obtained from the culture method, sequences were grouped using the CD_HIT program at 97% identity threshold [41,42]; we considered that the clustered OTUs were the same taxon. A representative OTU of each cluster was selected and compared with the NCBI and UNITE data using the BLAST algorithm [43].

### 2.5. Data Analysis

For the metabarcoding data, we estimated accumulation curves with and without singletons (OTUs and taxa that were only present in one sample) to evaluate the sampling effort and to compare the importance of rare taxa/OTUs between grasslands and tissue types. We calculated the OTU richness and Shannon and Simpson diversity indexes and we analyzed the effects of tissue and grassland type on fungal endophyte richness and diversity using two-way ANOVAs [49]. We calculated the relative abundance at the taxonomic level of phyla, orders, families and OTUs grouped into taxa using read sequences, within each tissue (shoot, rhizome and root) and each grassland type (LD and HD). We evaluated the effects of tissue and grassland type on fungal endophyte assemblages of *B. rupestre* using nonmetric multidimensional scaling (NMDS) with a Bray-Curtis dissimilarity index matrix, and we identified the distinctive fungal endophytes of a specific tissue and grassland type using indicator species tests [50], measuring the fidelity of the taxa to a particular situation [51].

## 3. Results

### 3.1. Comparison of B. rupestre Mycobiome Obtained by Culturing and Metabarcoding Methods

For the culture method, we obtained 28 isolates which were classified into a total of 19 morphotypes. Their corresponding sequences were matched in databases, a total of 15 taxa were obtained and classified to species (2), genus (9), family (3) and order (1) rank (Table 2). Ten taxa were isolated in plants collected in the LD grassland (66.6%), while eight were from the HD grassland (53.3%). We identified 2, 5 and 11 taxa from shoots, rhizomes and roots, respectively (Table 3).

The thirty samples of *B. rupestre* analyzed using the metabarcoding method produced 1,622,980 reads from 1822 OTUs before filtering and 513,671 reads from 352 OTUs after the filtering process. We obtained 316 OTUs from the LD grassland (61.1%) and 246 OTUs from the HD grassland (38.9%). There were 19,197 and 340 OTUs from shoots, rhizomes and roots, respectively. The OTU clustering process returned a total of 88 taxa: 38 assigned to genus, 16 to family, 19 to order, 9 to class and the remaining 6 to phylum or still unidentified (Appendix A). According to grassland type, 75 taxa were identified in the LD grassland (85.2%) and 52 in the HD grassland (59.1%). According to tissue type, 15, 37 and 82 taxa were identified in shoots, rhizomes and roots, respectively (Table 3).

The culturing method isolated 13 taxa out of 88 sequenced via metabarcoding. Since we used a conservative approach in the process of identification, it is likely that we arrived at different taxonomic levels of identification depending on the methodology, for example, *Codinaea* sp. (culturing) vs. Chaetosphaeriaceae (metabarcoding), Didymosphaeriaceae (culturing) vs. *Paracamarosporium* sp. (metabarcoding) and *Mollisia* sp. and *Phialocephala* sp. (culturing) vs. Mollisiaceae (metabarcoding). The rest of the isolated taxa did match at the taxonomic level assigned (*Albotricha* sp., *Drechslera* sp., *Falciphora* sp., Helotiaceae, *Lachnum* sp., *Microdochium phragmitis* and *Neoascochyta* sp.). Table 4 shows the complete information obtained from both methods for each sample, as well as the samples where the same taxon was isolated via the culturing method and also sequenced via the metabarcoding analysis. The two taxa isolated via culturing but not sequenced via metabarcoding were *Ilyonectria* sp. and *Omnidemptus graminis*. The latter was the most isolated fungal endophyte from shoots of *B. rupestre* using the culturing method.

Despite the remarkable differences in the number of sequences obtained using the two methods (28 isolates vs. 513,671 reads), the pattern of fungal endophyte richness and diversity among grassland and tissue types followed a similar trend, with the highest values in the root tissue and plants collected from the LD grassland.

### 3.2. The Mycobiome of B. rupestre According to the Metabarcoding Method

#### 3.2.1. Fungal Endophytic Richness and Diversity

The quantitative data from metabarcoding, based on read sequences, allowed an exhaustive characterization of the endophytic diversity of *B. rupestre*. Both the OTUs (352) and the clustering of OTUs into taxa (88) produced non-asymptotic species accumulation curves (Figure 2). However, 23 out of 88 taxa and 71 out of 352 OTUs were sequenced in only one sample (designated as singletons). Additional curves were constructed without singletons, suggesting that an increase in sampling effort would increase the number of rare taxa/OTUs but not the more common ones (Figure 2a,d). Accumulation curves comparing tissues and grassland types did not approach horizontal asymptotes (Figure 2b,c,e,f), therefore, greater sampling effort is required for reliable richness estimates.

The two factor ANOVA showed a significant effect of plant tissue (F = 19.9, *p* < 0.001) but not of grassland type (F = 2.5, *p* = 0.126) on OTU richness, whereas Shannon and Simpson indexes showed significant differences between grassland types (F = 5.1, *p* = 0.033 and F = 4.4, *p* = 0.046, respectively) and tissues (F = 32.7, *p* < 0.001 and F = 9.5, *p* < 0.001, respectively) (Figure 3).

#### 3.2.2. Taxonomic Assemblages for Grassland Types and Tissues

The relative abundance of phyla, orders and families was estimated from the read sequences. Most taxa were included in the phyla Ascomycota (71.21%) and Basidiomycota (21.21%). Figure 4 shows the relative abundance of orders and families according to tissue and grassland type.

Pleosporales dominated in shoots of plants from both grassland types (52.59% LD and 65.39% HD), followed by Phyllachorales (20.49%) and Pucciniales (16.05%) in the LD grassland and Hypocreales (17.45%) and Capnodiales (12.73%) in the HD grassland. The other orders did not exceed 4%, except Xylariales in the LD grassland (5.58%). Helotiales dominated in the belowground tissues of both grassland types ranging from 85.81% (Rhizome-HD) to 77.47% (Rhizome-LD), followed by Agaricales ranging from 20.93% (Rhizome-LD) to 7.17% (Root-LD). The other orders did not exceed 2.5% except Pleosporales in roots from the HD grassland (6.74%) (Figure 4, left).

Phaeosphaeriaceae dominated in shoots from both grassland types (63.48% HD and 34.08% LD), followed by unidentified family A (22.28%), Didymellaceae (16.94%) and Pucciniaceae (16.05%) in the LD grassland and unidentified family A (17.45%) and Mycosphaerellaceae (12.73%) in the HD grassland. The rest of the families did not exceed 5.5%. Hyaloscyphaceae dominated in belowground tissues ranging from 54.47% (Rhizome-HD) to 32.72% (Root-HD), followed by Helotiaceae ranging from 35.05% (Root-HD) to 13.16% (Rhizome-HD). Other families with relatively high abundance were Tricholomataceae and unidentified family A, ranging from 20.93% (Rhizome-LD) to 5.18% (Root-LD) and from 24.21% (Root-LD) to 8.18% (Root-HD), respectively. The other families did not exceed 4% (Figure 4, right).

The relative abundance of endophytic taxa after the OTU clustering process according to their high genetic similarity (97% threshold) was estimated from the read sequences. The most abundant read sequences were located in the root tissue and were reached by Helotiaceae (22.60%), *Lachnum* sp. A (21.94%), Helotiales A (8.29%) and *Albotricha* sp. A (7.00%). All of these were more abundant in plants collected in the LD grassland, with the exception of *Albotricha* sp. A.

In the roots, taxa with abundances higher than 5% were *Lachnum* sp. A (35.08%), Helotiaceae (24.51%) and Helotiales A (12.43%) in LD grassland plants and Helotiaceae (31.09%), *Albotricha* sp A (18.83%), *Lachnum* sp. A (12.50%), Agaricales A (9.65%) and Helotiales A (6.03%) in HD grassland plants (Table 5).

The dominant taxon in the shoots was Phaeosphaeriaceae (34.08% LD and 58.80% HD). In LD grasslands, it was accompanied by Phyllachorales (20.49%), *Puccinia* sp. (16.05%), *Neoascochyta* sp. A (14.94%) and *Microdochium* sp. (5.58%) and in HD grasslands by Sordariomycete A (17.45%), Mycosphaerellaceae (12.73%) and *Ophiosphaerella* sp. (4.68%). The remaining taxa did not exceed 4% (Table 6).

The dominant taxa in the rhizomes of the LD grassland were *Lachnum* sp A. (29.13%), Helotiaceae (25.95%), *Mycena* sp. A (20.88%), Helotiales A (10.98%) and *Albotricha* sp. A (5.23%) and in the rhizomes of the HD grassland *Albotricha* sp. A (50.58%), Helotiaceae (16.49%), *Lachnum* sp. A (9.01%), *Parasola* sp. (5.81%) and Tricholomataceae (4.21%) (Table 5).

#### 3.2.3. Indicator Species of the Fungal Assemblages

NMDS analysis showed that the fungal endophyte assemblage from above and belowground tissues of *B. rupestre* was clearly different (Figure 5a). In addition, fungal assemblages from root tissues (Figure 5d), unlike shoots (Figure 5b) and rhizomes (Figure 5c), displayed significant differences between grassland types.

The indicator species for shoot tissues were Phaeosphaeriaceae (*p* = 0.002), *Neoascohyta* sp. A (*p* = 0.005) and Phyllachorales (*p* = 0.029) and for root tissues were Helotiales A (*p* = 0.001), *Lachnum* sp. A (*p* = 0.001), Mollisiaceae A (*p* = 0.001), Pleosporales A (*p* = 0.001), *Drechslera* sp. (*p* = 0.001), *Lachnum* sp. B (*p* = 0.002), Agaricales A (*p* = 0.001), *Cladophialophora* sp. (*p* = 0.003), *Leohumicola* sp. (*p* = 0.003), *Pseudolachnella* sp. A (*p* = 0.007), Mollisiaceae C (*p* = 0.011), *Pseudolachnella* sp. B (*p* = 0.023), Unidentified B (*p* = 0.041), *Phragmocephala* sp. A (*p* = 0.019) and *Paracamarosporium* sp. (*p* = 0.036). No species were indicative of rhizome tissues.

The indicator species for grassland type were Phyllachorales (*p* = 0.044) and *Neoascochyta* sp. A (*p* = 0.035) in shoots collected in the LD grassland (Figure 5b). *Drechslera* sp. (*p* = 0.012) and Pleosporales A (*p* = 0.036) were indicators in roots from the HD grassland, while *Lachnum* sp. A (*p* = 0.007), *Phragmocephala* sp. A (*p* = 0. 042), *Paracamarosporium* sp. (*p* = 0.037) and *Pseudolachnella* sp. A (*p* = 0.047) were indicators in roots from the LD grassland (Figure 5d). No species in the rhizome tissues were indicative of plant community (Figure 5d).

## 4. Discussion

### 4.1. The Mycobiome of B. rupestre According to the Metabarcoding Data

The results of the metabarcoding showed that 88 taxa constituted the mycobiome of *B. rupestre* and that only seven taxa sequenced from the belowground tissues accounted for 81.2% of the total reads (Helotiaceae, *Lachnum* sp. A, *Albotricha* sp. A, Helotiales A, Agaricales A, *Mycena* sp. A and Mollisiaceae C), while the other 81 taxa were responsible for the remaining 18.8%, and 25 of them were only sequenced in a single sample. Therefore, a restricted sampling effort using the metabarcoding method was able to identify a small group of abundant fungal endophytes and a large group of rare species. The accumulation curves also supported the idea that extension of the sampling effort would enrich the group of rare species but not the most common species. This pattern of fungal endophyte distribution seems common to grasses [52] and indicates that a limited sampling effort is enough to provide good characterization of the dominant fungal species in plants, which is important considering the high cost of metabarcoding. However, when addressing studies on fungal richness and diversity, more extended sampling appears necessary to avoid an underestimation of the values.

The results of the study also highlight the importance of sampling the different tissues of plants to obtain a reliable characterization of its mycobiome [53,54]. Aboveground fungal assemblages were much poorer in species, less diverse and taxonomically different from those of rhizomes and roots, and this pattern was consistent between the grassland types, as observed by other authors in different plant species and different habitats [55,56,57]. The soil rhizosphere is the main route of fungal transmission to plants [58,59], and the high biomass of rhizome and roots developed by *B. rupestre* offers a large surface in contact with the soil microbiome. The majority of taxa identified were specific to a tissue, or exhibited a strong preference for it, and only five taxa appeared in all tissues (Helotiaceae, *Lachnum* sp. A, *Ophiosphaerella* sp., *Microdochium* sp. and *Epicoccum* sp.). As expected, the relative abundances of taxonomic orders and families also varied between tissues, with Pleosporales and Phaeosphaeriaceae more abundant in shoots and Helotiales and Hyaloscyphaceae more abundant in rhizomes and roots.

When comparing these results with previous characterizations of fungal endophyte assemblages in perennial temperate grasses based on culture techniques and extensive surveys, we realize the power of the metabarcoding tool, which is capable of identifying a large set of taxa with much less sampling effort. In *Dactylis glomerata*, 22 and 48 taxa were identified using culturing methods from the leaves and the roots of 120 samples [60], and in *Holcus lanatus*, 77 and 79 were identified in the same tissues of 77 samples [61]. The results of our survey of the leaves and roots of *B. rupestre* (2 and 11 taxa identified using the culturing method and 12 and 82 taxa identified using metabarcoding) obtained from a small number of samples in a regional sampling suggest that the real diversity and richness of the endophytic fungal assemblages of the previously studied grass species have probably been underestimated and would increase greatly if the novel metabarcoding techniques were used.

### 4.2. Culturing vs. Metabarcoding Methods

Modern massive sequencing techniques are gaining ground over traditional culturing methods due to the quantitative power of data that they are able to generate. With equal sampling effort, metabarcoding identified 13, 32 and 71 more taxa than culturing methods in shoots, rhizomes and roots, respectively, which means around ×5.8 times more species identified by the novel technique consistently in the three tissues. In similar studies comparing both methods, the metabarcoding identified ×5.2 and ×4.3 times more OTUs in roots of *Elymus repens* and *Deschampsia flexuosa* respectively than the culturing technique [62,63]. A parallel study using 240 plants of *B. rupestre* recognized 45 fungal endophytic taxa using the culturing method [39], in contrast to the 88 taxa sequenced using metabarcoding from 10 plants in the current survey. In this parallel study, the singletons isolated accounted for 48.9% of the taxa identified via culturing methods and 28.4% of the taxa identified via metabarcoding (with OTUs clustered with a 97% of similarity threshold).

Regarding belowground tissues, four fungal species with high incidence in root tissues were identified via both methodologies: *Albotricha* sp., Helotiaceae, *Lachnum* sp. and Mollisiaceae. In shoots, surprisingly, the most frequent shoot endophyte identified via the culturing method, *Omnidemptus graminis*, was not identified using the metabarcoding technique. *O. graminis* is a recently described taxon, included in a family associated with ongoing taxonomic changes due to molecular advances [64,65]. Its fast mycelial growth observed on culture plates may suggest the encrypting of other endophytes, but how *O. graminis* escaped the sequencing process of the metabarcoding is a matter that needs further study.

At this point, some issues need to be discussed when comparing the technical procedures of sequencing in both techniques. The ITS region is a universal and commonly used DNA barcode marker for fungi [66], and in the metabarcoding study undertaken by an external company, only the ITS2 region was amplified to identify the fungal sequences [67,68]. In the culturing method undertaken in the UPNA’s lab, the fungal mycelium was collected and the complete ITS region was amplified (ITS1-5.8S-ITS2), generating longer DNA sequences. We suggest that, since the ITS2 region is more restrictive, taxonomic inconsistencies may occur when short sequences are compared in the databases, thus affecting taxon identification [18]. The percentages of taxa identified for the metabarcoding were in the range 78.1–100%, and 97.6–100% for the culture sequencing, evidencing this restriction and indicating the value of sequencing the complete ITS region to achieve better fungal taxa identification. As a particular example, the taxon proposed as *Codinaea* sp. reached a match of 99.74% with the complete ITS region sequenced, while this percentage decreased to 97.52% when considering only the ITS2 region. As a consequence, the species was identified as Chaetosphaeriaceae in the metabarcoding, following a more conservative approach, although it was probably the same taxon. Similar situations may occur in other closely related taxa, when there is no reference specimen in the database [43,69]. Taxa identified as Mollisiaceae in our study probably belong to the genera *Mollisia* and/or *Phialocephala* [70,71] and the family Helotiaceae to the genera *Glarea* and/or *Hymenoscyphus* [72]. Both families were abundant in our samples. Other highly inclusive taxa, such as Pleosporales, raised similar doubts in the identification due to the still high uncertainty in the genetic characterization of the type specimens.

Despite the remarkable differences between the quantitative data generated using the two methods, the characteristics of the fungal assemblages in the different plant communities and tissues types are consistent between methods. Root tissues display the most diverse and rich fungal assemblages, and the endophytic community in plants collected in more disrupted, LD grasslands had the highest diversity and richness. Similar patterns have been reported in previous research in the area, conducted with a much greater sampling effort and using the culturing method [39], that analyzed the fungal assemblages in terms of the ecological mechanisms favored by the different disturbance regimes.

## 5. Conclusions

The endophytic mycobiome of *B. rupestre* is composed of a few abundant and many rare species, the identification of which depends on the sampling effort. Despite the restricted sampling effort, the two methodologies produced consistent results and detected the same trends in endophytic richness and diversity among tissues (roots > rhizomes > shoot) and grassland types (low-diversity > high-diversity). Comparatively, the metabarcoding method allowed the identification of a much larger number of taxa than the culturing method and revealed differences in richness and diversity that were not apparent with the culturing method (even when a larger number of samples was collected [39]).

Despite the promising results of the metabarcoding technique, the data indicate that a combination of the two methodologies is the best current option to obtain an adequate characterization of the plant fungal assemblage. In this study, metabarcoding did not identify *Omnidemptus graminis*, the most abundant fungal endophyte isolated in shoots via culturing; this recently described species is included in a family where there have been repeated taxonomic restructurings as a result of molecular advances [65].

## Figures and Tables

**Figure 1 biology-10-01246-f001:**
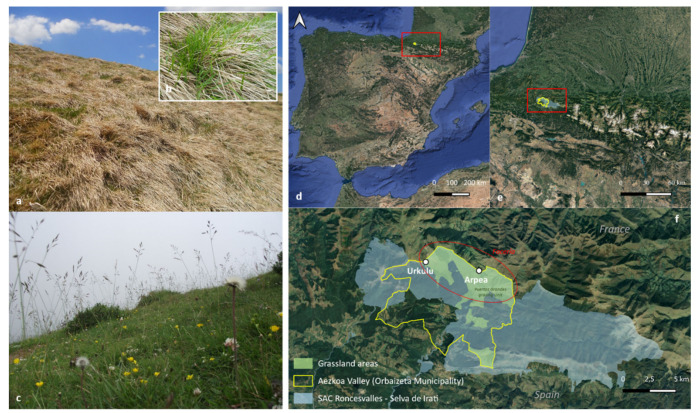
The appearance of low (**a**,**b**) and high (**c**) diversity grasslands. Location of the Aezkoa Valley in Spain (**d**) and within the western Pyrenees (**e**). The two locations (Arpea and Urkulu) where the samples were collected in the Roncesvalles-Selva de Irati SAC (**f**).

**Figure 2 biology-10-01246-f002:**
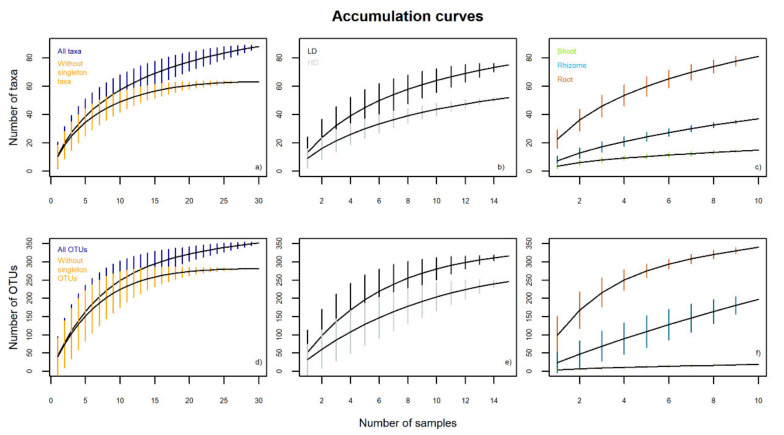
Taxon and OTU accumulation curves for the endophytic community of *B. rupestre* from metabarcoding (LD: low-diversity grassland, HD: high-diversity grassland). Black line shows the total number of taxa/OTUs, and vertical colored lines indicate the standard deviation.

**Figure 3 biology-10-01246-f003:**
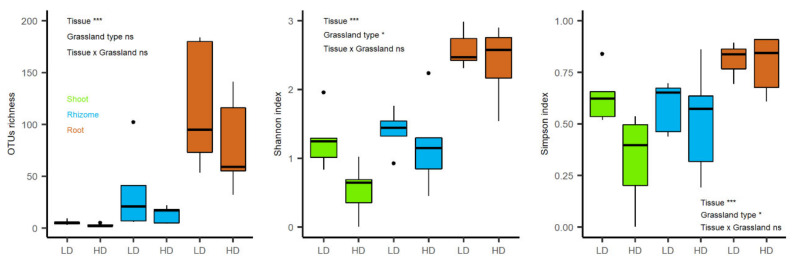
OTU richness and diversity indexes (Shannon and Simpson) for the endophytic community of *B. rupestre* from different tissues and grasslands (LD: low-diversity grassland, HD: high-diversity grassland). *** *p*-value < 0.001; * *p*-value < 0.05 and ns = no significance. Black points represent outliers.

**Figure 4 biology-10-01246-f004:**
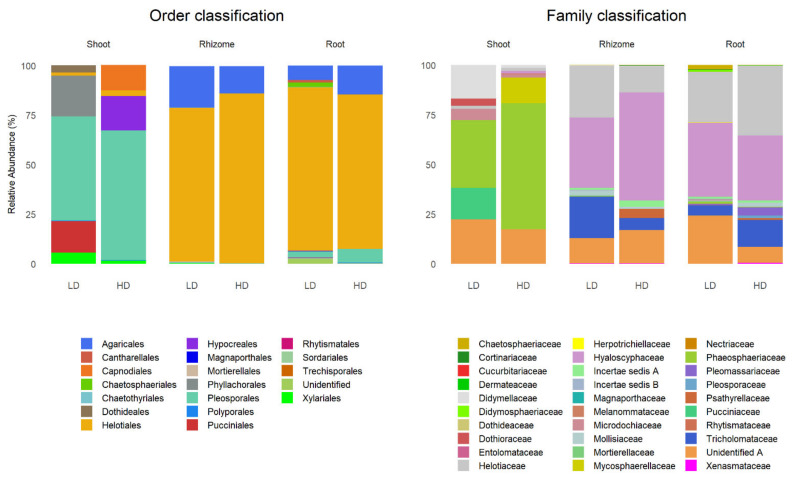
Taxonomic structure (orders, left and families, right) of fungal endophytes in *B. rupestre* tissues (shoot, rhizome and root) in the different grassland types (LD: low-diversity grassland, HD: high-diversity grassland).

**Figure 5 biology-10-01246-f005:**
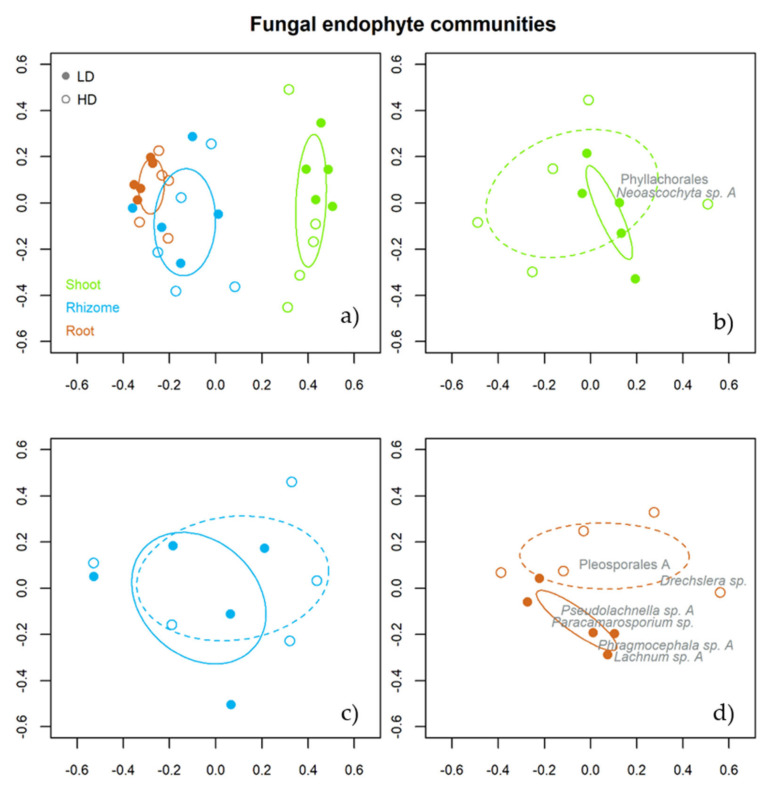
Non-metric multidimensional scaling analysis (NMDS) for the endophytic community of *B. rupestre* according to the effect of tissue (**a**) and plant diversity (shoot 1b, rhizome 1c & roots 1d). The ellipse formed by the solid line encompasses the fungal composition of *B. rupestre* tissues (**a**). The ellipses formed by broken and solid lines encompass the fungal composition of the low and high-diversity grassland, respectively (**b**–**d**). The taxon names in the graphs for shoots (**b**) and roots (**d**) are the indicator species for the effect of plant diversity.

**Table 1 biology-10-01246-t001:** General description of the study sites.

	Study Site	ARPEA	URKULU
	Type of Grassland	LD = Low Diversity	HD = High Diversity
General description	Location	−1°10′57″ W	–1°14′38″ W
		43°2′12″ N	43°2′49″ N
	Soil classification (WRB)	Cambic Umbrisol	Dystric Cambisol
	Altitude (m.a.s.l.)	893	1256
	Slope (%)	40	45
Management	Burning recurrence	High 1–2 years	Low 6–7 years
	Type of burning	Large grassland areas	Bush-to-bush
	Grazing level	Low to nonexistent	Moderate to high
	*B. rupestre* cover (%)	>80%	<25%

**Table 2 biology-10-01246-t002:** Fungal endophytes isolated from *B. rupestre* via the culturing method, their greatest percentage identity in both databases (NCBI and UNITE), the proposed taxon and the available accession number in GenBank.

	Match Taxon (NCBI)			Match Taxon (UNITE)			Taxon Proposed	GenBank Accession Number
		Accessio Number	Greatest Percentage Identity (%)		Accession Number	Greatest Percentage Identity (%)		
1	*Lachnellula hyalina*	NR_165202	90.11	*Albotricha* sp.	HM136666	98.22	*Albotricha* sp.	MW789554
2	*Codinaea paniculata*	NR_166297	99.74	*Codinaea* sp.	MT118230	99.74	*Codinaea* sp.	MW789567
3	*Paracamarosporium* sp.	NR_154318	94.28	*Paracamarosporium* sp.	MT882131	97.6	Didymosphaeriaceae	MW789559
4	*Drechslera* sp.	NR_153992	94.43	*Drechslera* sp.	UDB0174425	100	*Drechslera* sp	MW789560
5	*Falciphora oryzae*	NR_153972	96.69	*Falciphora* sp.	UDB0162916	99.76	*Falciphora* sp.	MW789558
6	*Glarea lozoyensis*	NR_137138	96.18	*Glarea* sp.	KF617491	99.58	Helotiaceae	MW789565
7	*Ilyonectria leucospermi*	NR_152889	99.36	*Ilyonectria crassa*	MT294410	100	*Ilyonectria* sp.	MW789566
8	*Lachnellula hualina*	NR_165202	88.89	*Lachnum virgineum*	MT133783	98.15	*Lachnum* sp.	MW789564
9	*Microdochium phragmitis*	NR_132916	100	*Microdochium phragmitis*	MH861162	100	*Microdochium phragmitis*	MW789562
10	*Mollisia asteliae*	NR_173037	96.44	*Mollisia* sp.	KJ188683	98.69	*Mollisia* sp.	MW789555
11	*Phialocephala spaheroides*	NR_121302	95.71	*Loramyces* sp.	KF618060	99.36	Mollisiaceae	MW789556
12	*Neoascochyta dactylidis*	NR_170041	100	*Neoascochyta* sp.	MT185527	100	*Neoascochyta* sp.	MW789561
13	*Omnidemptus graminis*	NR_164058	100	*Omnidemptus graminis*	MK487758	100	*Omnidemptus graminis*	MW789553
14	*Phialocephala sphaeroides*	NR_121302	89	*Phialocephala* sp.	JN995646	98.87	*Phialocephala* sp.	MW789563
15	*Paraphaeosphaeria michotii*	NR_155640	91.41	Pleosporales	MN450621	100	Pleosporales	MW789557

**Table 3 biology-10-01246-t003:** Total number of reads, OTUs and taxa associated with *B. rupestre* tissues and the type of grassland where plants were collected (LD: low-diversity grassland, HD: high-diversity grassland).

			Type of Grassland		Tissue		
					Shoot	Rhizome	Root
**Metabarcoding method**		**Reads**	**LD**	313,621	4680	47,268	261,673
			**HD**	200,050	3204	29,692	167,154
			**Total**	513,671	7884	76,960	428,827
		**OTUs**	**LD**	316	12	165	305
			**HD**	246	11	58	236
			**Total**	352	19	197	340
		**Taxa**	**LD**	75	10	27	69
			**HD**	52	10	23	45
			**Total**	88	15	37	82
**Culture method**		**Taxa**	**LD**	10	2	3	6
			**HD**	8	1	3	5
			**Total**	15	2	5	11

**Table 4 biology-10-01246-t004:** Culturing and metabarcoding comparison. Total number of reads, OTUs and taxa and match identification for each sample.

Sample		Culture Method	Match Methods	Metabarcoding		
		Isolated Taxa		Taxa	OTUs	Reads
Shoot LD	1	*Neoascochyta* sp.	✓	5	5	233
	2	*Omnidemptus graminis*	×	5	6	1639
	3	*Omnidemptus graminis*	×	7	9	1619
	4	*Omnidemptus graminis*	×	3	3	207
	5		×	4	4	982
Shoot HD	6		×	2	2	229
	7	*Omnidemptus graminis*	×	2	2	37
	8	*Omnidemptus graminis*	×	5	5	644
	9	*Omnidemptus graminis*	×	1	1	13
	10		×	3	3	2281
Rhizome LD	1		×	8	102	12,546
	2		×	11	21	24,377
	3	Didymosphaeriaceae	×	4	6	1312
		Helotiaceae	✓			
	4		×	5	7	831
	5	Mollisiaceae	✓	12	41	8202
Rhizome HD	6	Helotiaceae	✓	6	22	5621
	7	Helotiaceae	✓	3	4	267
	8	*Phialocephala* sp.	×	11	17	15,380
	9		×	4	5	2035
	10	*Microdochium phragmitis*	×	11	18	6389
Root LD	1	Didymosphaeriaceae	×	15	180	49,606
	2	*Falciphora* sp.	✓	29	53	40,482
		*Codinaea* sp.	×			
	3	Didymosphaeriaceae	×	27	184	70,132
	4	*Mollisia* sp.	×	34	73	52,335
	5	Pleosporales	×	31	95	49,118
		Didymosphaeriaceae	×			
		*Lachnum* sp.	✓			
Root HD	6	Helotiaceae	✓	18	141	62,044
	7	Mollisiaceae	✓	20	116	23,703
	8	*Albotricha* sp.	✓	16	32	12,379
	9	*Albotricha* sp.	✓	20	55	47,814
	10	*Drechslera* sp.	✓	17	59	21,214
		*Ilyonectria* sp.	×			

**Table 5 biology-10-01246-t005:** List of the most abundant taxa in *B. rupestre* underground tissues. The relative abundance is based on number of reads, number of OTUs and infected plants (out of five). Shaded taxa were sequenced in both underground tissues. The complete table is available in Appendix B.

	ROOT								RHIZOME							
Endophyte Taxon	Relative Abundance (%)		Reads		OTUs		Infected Plants		Relative Abundance (%)		Reads		OTUs		Infected Plants	
	LD	HD	LD	HD	LD	HD	LD	HD	LD	HD	LD	HD	LD	HD	LD	HD
**Helotiaceae**	24.51	31.09	64,132	51,968	114	116	2	4	25.95	16.49	12,267	4897	94	18	2	4
***Lachnum* sp. A**	35.08	12.5	91,790	20,889	36	20	5	5	29.13	9.01	13,771	2676	5	7	4	4
***Albotricha* sp. A**	1.71	18.83	4465	31,473	7	6	3	3	5.23	50.58	2473	15,018	4	6	3	3
**Helotiales A**	12.43	6.03	32,534	10,072	40	28	5	5	10.98	2.77	5188	823	25	1	2	3
**Agaricales A**	3.55	9.65	9281	16,124	3	3	2	4	0.05	0	24	0	2	0	1	0
***Mycena* sp. A**	2.03	0.17	5323	289	10	2	4	1	20.88	0	9870	0	1	0	1	0
**Mollisiaceae C**	4.17	0.45	10,913	745	2	1	4	1	0.42	0.04	198	13	2	1	1	1
**Pleosporales A**	0.56	4.04	1476	6751	2	4	3	5	0	0.04	0	12	0	1	0	1
***Glarea* sp.**	0.41	3.94	1060	6589	2	1	2	1	0	0.29	0	86	0	1	0	1
**Mollisiaceae B**	0.43	1.89	1118	3161	1	3	2	3	2.35	1.44	1111	429	1	3	2	2
**Mollisiaceae D**	0.89	1.07	2330	1782	1	2	2	1	0.78	3.56	369	1056	1	2	2	2
**Chaetosphaeriaceae**	1.76	0	4608	0	4	0	1	0	0.07	0	33	0	1	0	1	0
***Mycena* sp. B**	0	2.08	0	3479	0	3	0	1	0	3.34	0	993	0	1	0	1
**Tricholomataceae B**	0	1.48	0	2474	0	1	0	2	0	4.21	0	1251	0	1	0	1
***Lachnum* sp. B**	0.38	1.27	1007	2119	11	7	4	4	0.87	0.14	411	43	1	1	1	1
**Cantharellales**	1.3	0	3397	0	2	0	1	0								
***Parasola* sp.**	0	0.9	0	1503	0	1	0	1	0	5.81	0	1725	0	3	0	1
**Unidentified A**	1.21	0.01	3174	19	2	1	1	1								
***Ophiosphaerella* sp.**	0.96	0.32	2513	535	2	1	2	1	0.11	0.06	50	17	1	1	1	1
**Mollisiaceae A**	0.88	0.4	2309	666	4	3	4	5	0.03	0.28	13	83	1	1	1	1
***Drechslera* sp.**	0.03	1.43	87	2388	2	2	2	5								

**Table 6 biology-10-01246-t006:** List of taxa in the *B. rupestre* shoots and their relative abundance based on number of reads, number of OTUs and infected plants (out of five).

SHOOT								
Endophyte Taxon	Relative Abundance (%)		Reads		OTUs		Infected Plants	
	LD	HD	LD	HD	LD	HD	LD	HD
Phaeosphaeriaceae	34.08	58.80	1595	1884	2	1	4	2
Phyllachorales	20.49	0	959	0	1	0	4	0
*Puccinia* sp.	16.05	0	751	0	3	0	1	0
*Neoascochyta* sp. A	14.94	0.53	699	17	1	1	4	1
Sordariomycetes A	0	17.45	0	559	0	1	0	1
Mycosphaerellaceae	0	12.73	0	408	1	0	0	1
*Microdochium* sp.	5.58	1.59	261	51	1	1	4	1
Dothideales	3.65	0	171	0	1	0	2	0
*Ophiosphaerella* sp.	0	4.68	0	150	0	1	0	1
*Epicoccum* sp.	2.01	0.75	94	24	1	1	1	1
Helotiaceae	1.41	1.56	66	50	1	2	2	3
*Periconia* sp.	1.56	0	73	0	1	0	1	0
*Lachnum* sp. A	0	1.28	0	41	0	1	0	1
*Phragmocephala* sp. B	0	0.63	0	20	1	0	0	1
Unidentified C	0.23	0	11	0	1	0	1	0
	100	100	4680	3204				

## Data Availability

The sequencing data have been deposited in GenBank at NCBI with their accession number.

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
