# Peer review of "Comparison of Culturing and Metabarcoding Methods to Describe the Fungal Endophytic Assemblage of Brachypodium rupestre Growing in a Range of Anthropized Disturbance Regimes"

_biology, 2021, doi:10.3390/biology10121246_

Round 1
Reviewer 1 Report
This ms present a comparison of methods (culturing versus metabarcoding) to analyse the fungal endophytic assemblage of Brachypodium rupestre growing in mountain grasslands with two different levels of plant diversity, and condsidering different types of plant tissues (root, rhizome and shoot). Although, in general, the results are not surprising, it is very interesting and sound the comparison of the results of both methods. The significance of the content and the quality of the presentation is very good.
The differences in number between both methods, could be partly due to the sampling? I mean, did you select the plant fragments for isolation and metabarcoding analysis in the same way?
Lines 121: do you mean -80ºC?
Line 127: instead of mini Petri dish, use the diameter of the Petri plate (and also in line 123)
Lines 115-121: Please separate the symbol from the number
Lines 189-194: many data are already in tables. Perhaps it is more interesting to focus here on the OTUs numbers
Lines 213: do you mean 28 ‘isolates’??
Line 238: change to ‘… effect of plant tissue’
Line 285: table 6 is mentioned in the text before table 5
Lines 282-285: Why did you chose the value of 5% of abundance?
Figure 4. It is suggested to increase the size of the symbols to distinguish colours. It is really hard to see some of them. Perhaps you can merge those with an abundance lower than 5% in a group called ‘others’.
Table 5, do you mean ‘infected plants’ (instead of infected samples)?
Figure 5: Label each graph with the corresponding letter (a, b, c, d)
Add the title of tables appendix A and B
Discussion
It would be interesting to include in the discussion some other references in which both methods are compared.
Lines 364-371: the reference and results presented here for other grass species such as Holcus lanatus show the identification of a very high number of taxa by culture techniques, which is similar to the number obtained in this study for Brachypodium by means of metabarcoding. Is it an effect of differences or failures in your culture method? Or it shows an high diversity in Holcus, which could be even increased by metabarcoding method?
Lines 373-374. I don´t think that opponent is the right word. I think that are different methods that could be used depending of the objective of the research. For fungal diversity metabarcoding is more adequate, but if you want to obtain isolates for other purposes, the you need of culture methods. Your results also show that some abundant fungal isolates were not identified by sequencing, which is a bit disappointing
Lines 416-417: do you think that both data are really comparable at the same level?
Author Response
We are deeply grateful for the positive evaluation and constructive comments from the reviewer 1. We have followed their suggestions and also have added references in relation to the comparison studies of both methods.

Reviewer 2 Report
Here is the review of the paper entitled "Comparison of culturing and metabarcoding methods to describe the fungal endophytic assemblage of Brachypodium rupestre growing in a range of anthropized disturbance regimes" written by María Durán & co-authors.
The aim of the study is to analyze the endophytic fungal community in shoots, rhizomes and roots of the grass Brachypodium rupestre from a natural grasslands in Spain. The methods used for assessment of fungal community were isolation of fungal cultures coupled with molecular identification (based on Sanger sequencing - ITS marker gene) and eDNA metabarcoding (based on ITS2 marker gene region). Both methods detected the same situation regarding the richness and fungal diversity among the diiferent plant tissues and grasslands. The metabarcoding method identified almost six times more taxa than the traditional isolation from pure cultures. At the same time the most common endophtye identified by classic Sanger sequencing of fungal DNA barcode region (Omnidemptus graminis) was not detected by eDNA metabarcoding. Conclusion of the study was that the combination of both methods, and different plant parts (shoots, rhizomes, and roots) is preferred for detection of the complete fungal endophytic community in plant species analyzed.
The paper needs to be substantially improved to be published in Biology journal. English language require moderate changes since the some parts of the text are not easily understandable. In Introduction section, the plant species analyzed for fungal endophytic community (Brachypodium rupestre) is not properly described (e.g. importance for humans, ecology - habitat requirements, previous studies on endophytic fungi in B. rupestre etc.). Methods section should be improved as well and there are some unclear/unprecise sentences in the other sections (marked in pdf attached). It seems that metabarcoding procedure is properly executed and well explained in the text. Statistical methods are appropriate. Finally, please go through my remarks in pdf manuscript file and make needed corrections.
The paper should be reconsidered after the major revision.
Best,
Reviewer

Author Response
We very appreciate the comments and the intensive review of the second reviewer. We agree with all suggestions and we have tried to improve the understanding of the highlighted sentences. Small mistakes have been modified and are not mentioned in this document (masl -> m asl; 2cm -> 2 cm; Italic font; Sordariomycete -> Sordariomycetes, etc).

Round 2
Reviewer 2 Report
The authors substantially improved the manuscript according to my instructions and it could be accepted for publication in Biology journal.
Best, reviewer